# Enhanced *β*-adrenergic response in mice with dominant-negative expression of the PKD2L1 channel

**Manabu Murakami**[1]*, **Agnieszka M. Murakami**[2], **Takayuki Nemoto**[1], **Takayoshi Ohba**[3], **Manabu Yonekura**[4], **Yuichi Toyama**[4], **Hirofumi Tomita**[4], **Yasushi Matsuzaki**[5], **Daisuke Sawamura**[5], **Kazuyoshi Hirota**[6], **Shirou Itagaki**[7], **Yujiro Asada**[8], **Ichiro Miyoshi**[9]

**1** Department of Pharmacology, Faculty of Medicine, University of Miyazaki, Miyazaki, Miyazaki, Japan,
**2** Department of Pharmacology, Hirosaki University Graduate School of Medicine, Hirosaki, Aomori, Japan,
**3** Department of Cell Physiology, Akita University School of Medicine, Akita, Akita, Japan, **4** Department of Cardiology and Nephrology, Hirosaki University Graduate School of Medicine, Hirosaki, Aomori, Japan,
**5** Department of Dermatology, Hirosaki University Graduate School of Medicine, Hirosaki, Aomori, Japan,
**6** Department of Anesthesiology, Hirosaki University Graduate School of Medicine, Hirosaki, Aomori, Japan,
**7** Collaboration Center for Community and Industry, Sapporo Medical University, Sapporo, Hokkaido, Japan,
**8** Division of Pathophysiology, Department of Pathology, Faculty of Medicine, University of Miyazaki, Miyazaki, Miyazaki, Japan, **9** Department of Laboratory Animal Medicine, Tohoku University School of Medicine, Sendai, Miyagi, Japan

* mmura0123@hotmail.co.jp

**Data Availability Statement:** All relevant data are within the manuscript and its Supporting Information files.

## Abstract

Polycystic kidney disease (PKD) is the most common genetic cause of kidney failure in humans. Among the various PKD-related molecules, PKD2L1 forms cation channels, but its physiological importance is obscure. In the present study, we established a transgenic mouse line by overexpressing the dominant-negative form of the mouse PKD2L1 gene (i.e., lacking the pore-forming domain). The resulting PKD2L1del-Tg mice exhibited supraventricular premature contraction, as well as enhanced sensitivity to β-adrenergic stimulation and unstable R-R intervals in electrocardiography. During spontaneous atrial contraction, PKD2L1del-Tg atria showed enhanced sensitivity to isoproterenol, norepinephrine, and epinephrine. Action potential recording revealed a shortened action potential duration in PKD2L1del-Tg atria in response to isoproterenol. These findings indicated increased adrenergic sensitivity in PKD2L1del-Tg mice, suggesting that PKD2L1 is involved in sympathetic regulation.

## Introduction

Polycystic kidney disease (PKD) is a common autosomal dominant Mendelian disorder affecting 0.1% of the population, and it is the most common genetic cause of kidney failure in humans [1]. PKD is a monogenic hereditary disease characterized by multiple, progressive bilateral cystic dilatations of renal tubules, which result in renal failure. Positional cloning analysis has revealed two loci of etiological importance: polycystin-1 (PKD1), mapped to chromosome 16p13.3, and polycystin-2 (PKD2), mapped to chromosome 4q21–23 [2–5]. Ninety-

**Funding:** This research was sponsored, in part, by Grants-in-Aid for Scientific Research from JSPS, KAKENHI (17K08527, 17H04319, 16K09489, and 20K07255). No additional external funding was received for this study.

**Competing interests:** The authors have declared that no competing interests exist.

five percent of patients with autosomal-dominant polycystic kidney disease (ADPKD) exhibit mutations in PKD1 (about 85%) and PKD2 (about 15%). Notably, cardiovascular complications are the major causes of morbidity and mortality in patients with PKD. Cardiac-related death is estimated to be 1.6- to 3.2-fold more frequent in these patients than in the general population [6].

PKD1 and PKD2 participate in the transient receptor potential polycystic (TRPP) channel subfamily. There are two types of PKD-related TRPP genes: PKD1-like (PKD1, also known as TRPP1) and PKD2-like (TRPP2, TRPP3, and TRPP5). As an important PKD2-like gene, PKD2 (TRPP2) forms a calcium-permeable nonselective cation channel [7–9]. Four other forms of PKD-related genes have been identified by means of computer-based homology screening [10]. Among them, polycystin-2L1 (PKD2L1, also known as TRPP3 or PKDL) has high homology to PKD2. Mice lacking PKD2L1 and Pax2 genes develop lethal PKD [11], while other homologous-recombinant gene-targeting approaches revealed an epileptic phenotype in PKD2L1-null mice [12]. In previous studies, our group demonstrated that PKD2L1 forms cation channels [13]. PKD2L1 is also involved in high-salt-induced cardiac hypertrophy [14]. PKD2L1, is a non-selective cation channel that is permeable to $Ca^{2+}$, $K^+$, and $Na^+$ [12, 15]. This channel is expressed in the kidney, brain, heart, and other tissues [13, 15–19]. PKD2L1 forms cation channels with PKD1L3 interacting with its C-terminal domain or its transmembrane domain [20, 21].

In a previous study, Yao et al. demonstrated an epileptic phenotype in PKD2L1-deficient mice, along with interaction and co-localization of PKD2L1 with β2-adrenergic receptors (β2ARs) in the brain [12]. PKD2L1 deficiency led to the loss of β2AR on neuronal cilia, which was accompanied by a 27% reduction in cyclic adenosine monophosphate (cAMP) levels in the central nervous system. Thus far, multiple aspects of PKD2L1 remain unclear, such as the physiological importance of PKD2L1 in channel formation in cardiac tissue and the importance of PKD2L1 in PKD. Here, to further characterize the roles of PKD2L1, we constructed a dominant-negative form of the PKD2L1 gene and overexpressed it in transgenic (Tg) mice.

## Methods

### Ethics statement

This study was performed with the approval of the Institutional Review Board of Miyazaki University. All experiments were conducted in accordance with the Guidelines for the Use of Laboratory Animals at Miyazaki University. The animals were housed throughout the study under a constant 12-h light/dark cycle with free access to food and water. Mice aged 12–20 weeks were used in all experiments. All procedures were performed under general anesthesia as described below, and all efforts were made to minimize suffering. To isolate the heart, the mice were anesthetized with an intraperitoneal injection of a mixture of medetomidine hydrochloride (0.315 mg/kg), midazolam (2.0 mg/kg), and butorphanol tartrate (2.5 mg/kg). To prevent any pain associated with the injections, the mice were first anesthetized by inhaling 80% carbon dioxide and 20% oxygen before the intraperitoneal injection.

### Generation of dominant-negative PKD2L1 Tg mice

Mouse PKD2L1 cDNA was used as reported previously [13]. Two PCR fragments were amplified using specific primers, and three restriction sites (BglII, XbaI, and NotI) were introduced to construct the DNA. The primers BglII-PKD2L1For (5′-AAAAGATCTATGAATAGTATGG AAAGCCCC-3′) and mPKD2L1AXbaIRev (5′-AAATCTAGAGGTGGAGGAGAGCTGTGTCA TGG-3′) were used to amplify the PKD2L1 sequence from the start codon to the fourth transmembrane domain. mPKD2L1BXbaIFor (5′-AAATCTAGAAACGACACATACTCCGAGGT

CAAG-3′) and NotI-PKD2L1Rev (5′-AAGCGGCCGCACAACTCCTTCCAGAACACA-3′) were used to amplify the 3' terminal sequences of the construct. Two fragments were ligated at the XbaI sites and inserted into multiple cloning sites (BamHI and NotI) of the pQBI expression plasmid (Wako Chemicals, Osaka, Japan), which contained the cytomegalovirus (CMV) promoter, enhanced green fluorescent protein, and bovine growth hormone poly(A) sequences. This cDNA (PKD2L1del) contained the first four transmembrane domains of the PKD2L1 gene and no pore-forming fifth and sixth transmembrane domains, which are essential for cation channel formation. To evaluate the functions of the pore-forming domains, other sequences were maintained as in the original mouse PKD2L1 cDNA. PKD2L1del-Tg mice were generated as described previously [22].

## RNA isolation and reverse-transcription (RT)-PCR analysis

Total RNA was isolated from the hearts and kidneys of C57/BL6 and PKD2L1del-Tg mice using the RNeasy kit (Qiagen Inc., Valencia, CA, USA). The reverse-transcription reaction was performed using a first-strand cDNA synthesis kit (Invitrogen, Carlsbad, CA, USA) in a volume of 25 μL at 42˚C for 45 min.

PKD2L1-specific sequences were amplified for 39 cycles using specific primers (PKD2L1-5Terminal, S1 Table in S1 File). Comparative RT-PCRs were performed under the same conditions for 34 PCR cycles. Control RT-PCRs were performed for 28 cycles using β-actin-specific primers. Subsequently, specific sequences of β1-adrenergic receptors, β2ARs, and muscarinic 2 receptors were amplified by PCR (37 cycles). β-actin sequences were amplified by PCR as an internal control (35 cycles). The primers used for the RT-PCR analysis are shown in S1 Table in S1 File.

## Western blot analysis

Protein samples from the heart were homogenized and lysed in 50 mM Tris (pH 7.5), 140 mM NaCl, and 5 mM EDTA with a protease inhibitor cocktail (Complete Mini; Sigma-Aldrich, St. Louis, MO, USA). Aliquots (100 μg) of the homogenate from each mouse were resolved on 7.5% sodium dodecyl sulfate-polyacrylamide gel electrophoresis and subjected to western blotting. Commercially available polyclonal antibodies specific for PKD2L1 (Novus Biologicals, Centennial, CO, USA). An anti-glyceraldehyde-3-phosphate dehydrogenase (GAPDH) antibody was used as a control.

## General anesthesia

Mice (12–16 weeks of age) were anesthetized in an induction chamber ($25 \times 25 \times 14$ cm) containing 4% isoflurane (Forane; Abbott Japan Co., Ltd., Tokyo, Japan) and room air. Anesthesia was maintained for 45 min (anesthetic maintenance state) using 2% isoflurane inhaled anesthesia at an airflow rate of 0.5 L/min. Ten minutes after inducing anesthesia, baseline electrocardiogram and echocardiography recordings were performed for 5 min, followed by pharmacological tests. All experiments were conducted between 10:00 and 16:00 [23].

## Electrocardiography

Electrocardiography (ECG) was performed as described previously [23]. Analog ECG signals were transferred to a receiver, digitized using the Power Lab system, and analyzed using LabChart 8 software (AD Instruments, Dunedin, New Zealand). The heart rate and other ECG parameters were evaluated using commercial software (ML846 Power Lab system; AD Instruments). Heart rate variability (HRV) was determined according to the normal-to-normal R-R

interval (SDNN), which is presumed to reflect the integrity of cardiac vagal control. The HRV analysis featured a time domain analysis (based on calculating the standard deviation of the SDNN) and an analysis of the HRV power spectrum. The SDNN represents total variability. The HRV power spectrum has three components: very low frequency, low frequency (LF), and high frequency (HF). The LF component reflects sympathetic/parasympathetic tone, while the HF component reflects parasympathetic tone. We defined the ranges of the spectral components as $< 0.15$, very low frequency; $> 0.15$ and $< 1.5$, LF; and $> 1.5$ and $< 5$, HF, according to the manufacturer's instructions. The mice were administered atropine (1.0 mg/kg) to induce parasympathetic blockade or propranolol (1.0 mg/kg) to induce sympathetic blockade for the pharmacological analysis.

### Inotropic atrial contraction

Left atria were dissected free of ventricular tissue and placed in an oxygenated 37˚C tissue bath containing Tyrode's solution (123.8 mM NaCl, 5.0 mM KCl, 2.0 mM $CaCl_2$, 1.2 mM $MgCl_2$, 25.0 mM $NaHCO_3$, and 11.2 mM glucose).

Isolated left atria were stimulated at 2 Hz and a voltage immediately above the threshold for 30 min of basal contraction (1-ms duration) to stabilize basal contraction. The isometric contractile force was measured using a force transducer (CD200; Nihon Koden, Tokyo, Japan) [24].

### Action potentials

Isolated right atria were mounted in an organ bath (10 mL) continuously perfused with oxygenated Tyrode's solution containing the following (in mM): NaCl 126.7, KCl 5.4, $CaCl_2$ 1.8, $MgCl_2$ 1.05, $NaHCO_3$ 22, $NaHPO_4$ 0.42, and glucose 5.6. After equilibration with room air, the pH of the solution was 7.4 (25˚C). Action potentials were recorded from the external epicardial surface using conventional glass pipettes filled with a 3.0 M KCl solution (tip resistance, 20–30 MΩ). Action potentials were recorded using an amplifier (MEZ-8301; Nihon Koden) and stored on a personal computer for subsequent off-line analyses of the following parameters: resting membrane potential, action potential amplitude, maximum up-stroke velocity (dV/dtmax), and action potential durations at 40% and 90% repolarization. All data were acquired and analyses were performed using LabChart software (AD Instruments). A $Ca^{2+}$ antagonist (verapamil) was added to Tyrode's solution from concentrated stock solutions to yield the final concentration (10 μM) for the pharmacological evaluation of verapamil.

### Statistical analysis

Results are expressed as the mean ± standard error. Means were compared using the Newman–Keuls *post-hoc* multiple-range test. One-way analysis of variance (ANOVA) was used to detect differences followed by Dunnett's *t*-test. The isoproterenol pharmacological analysis (ECG, contraction, and action potential duration) was evaluated with two-way ANOVA followed by the Bonferroni comparison, and *p*-values $< 0.05$ were considered significant.

## Results

### Establishment of dominant-negative PKD2L1-overexpressing mice

The overexpression construct was generated by ligating two PCR-amplified fragments of the PKD2L1 coding sequence into the pQBI plasmid (Wako Chemicals), controlled by the CMV promoter; enhanced green fluorescent protein sequence was followed by the PKD2L1 sequence and a rabbit globin poly(A) signal (Fig 1A). This "PKD2L1del" cDNA lacks the fifth

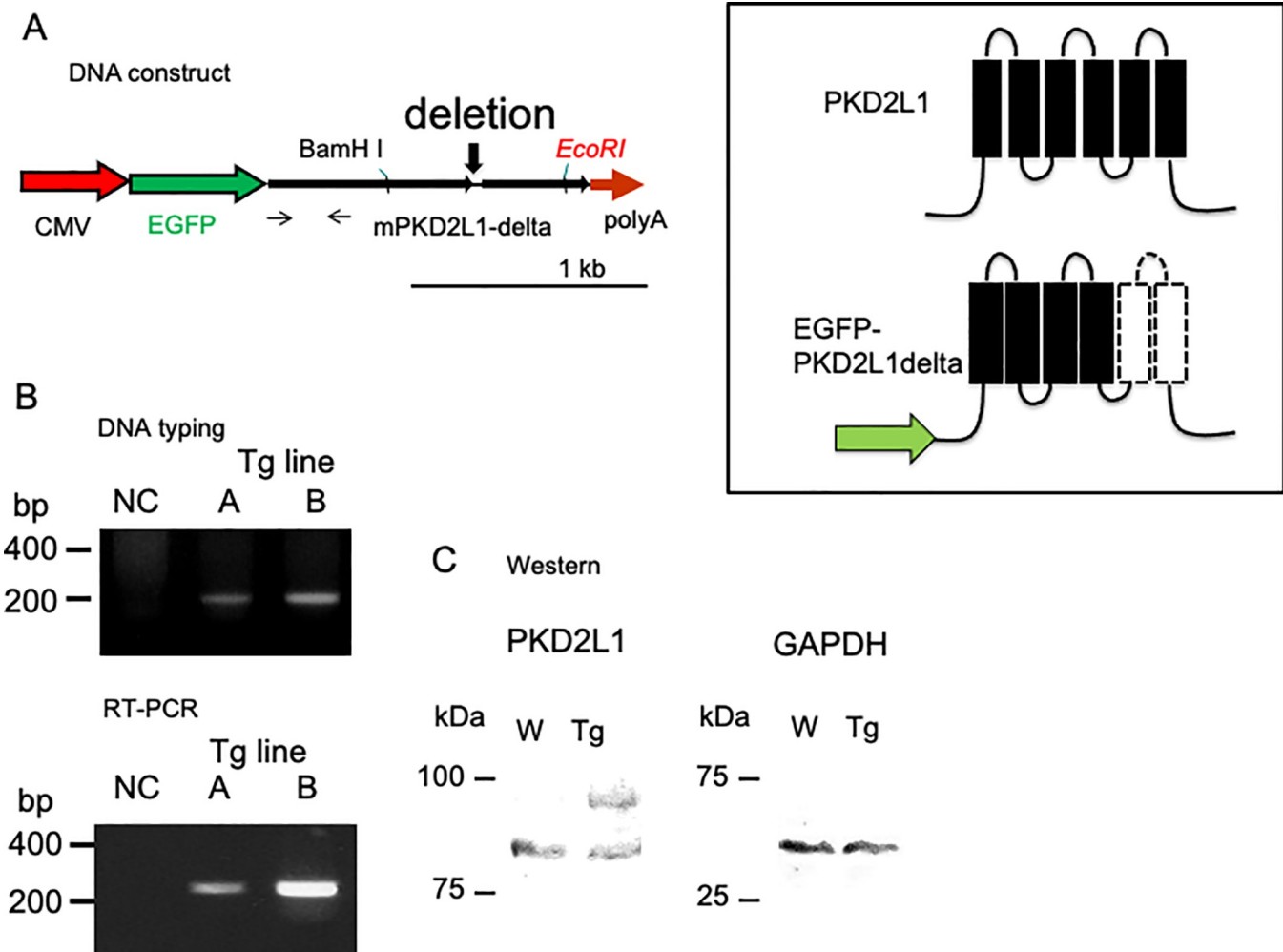

**Fig 1.** A) Molecular characterization of PKD2L1del overexpressed in Tg mice. Schematic of the CMV promoter/PKD2L1del construct used for the generation of PKD2L1del-Tg mice. The deletion site is also shown. The primer sets for each PCR amplification (DNA typing and RT-PCR) are indicated with arrows. The inset shows predicted schematic transmembrane constructs of full-length PKD2L1 (upper panel) and PKD2L1del. Scale bar = 1 kb. B) DNA typing of Tg mice. Tg-A mice contained a single copy of the external gene, whereas mice in Tg-B contained two copies. RT-PCR analysis of cardiac tissue from Tg mice. Tg-B mice showed enhanced PKD2L1 expression in the heart. *NC*, negative control without cDNA. The names of the Tg lines (A and B) are indicated. C) Western blot analysis of the hearts from the WT and Tg mice. Representative immunoblots of membranes from WT and Tg mice analyzed for the expression of PKD2L1 (left panel) and GAPDH (right panel).

and sixth pore-forming transmembrane domains, which are presumed to be essential for cation channel formation (Fig 1A inset). All other sequences were maintained as in the wild-type (WT) mouse PKD2L1 cDNA. The plasmid insert (from the promoter to the poly(A) sequence) was prepared and injected into the pronucleus [22].

Tg offspring were genotyped by quantitative PCR analysis, confirming the generation of two transgenic lines (Tg-A and Tg-B) with germ-line transmission (Fig 1B, DNA typing, lanes A and B: Tg-A and Tg-B, respectively). The Tg-B line was estimated to carry two copies of the transgene, compared with the Tg-A line, which had a single copy of the transgene (Fig 1B, RT-PCR, S3 Fig in S1 File).

The RT-PCR analysis revealed expression of the Tg-A and Tg-B transgenes in the heart (Fig 1B, RT-PCR, lanes A and B: Tg-A and Tg-B, respectively). Because no significant phenotype was observed and expression of the transgenes was high, the Tg-B mouse line was used for further analysis.

To further investigate the expression of the transgene at the protein level, we analyzed the protein expression of PKD2L1 in the heart of WT and Tg-B mice by immunoblotting (Fig 1C). The endogenous PKD2L1 formed a single 85-kDa band in the WT control mice (Fig 1C). The Tg mice showed an additional 95-kDa band, indicating overexpression of the dominant-negative form of the PKD2L1 transgene with the CMV promoter (Tg). Anti-GAPDH antibody binding (Fig 1C, GAPDH) confirmed comparable loading of proteins on the gel. The histological examination revealed no obvious changes in the heart or kidneys.

### Expression profile using RT-PCR

To investigate the influence of overexpression of dominant-negative PKD2L1, we analyzed the expression of PKD-related genes in the kidneys and hearts of Tg and WT control mice by RT-PCR (Fig 2A). Using the CMV promoter combined with endogenous expression, PKD2L1 was significantly overexpressed both in kidney and heart (S3 Fig in S1 File). The mRNA

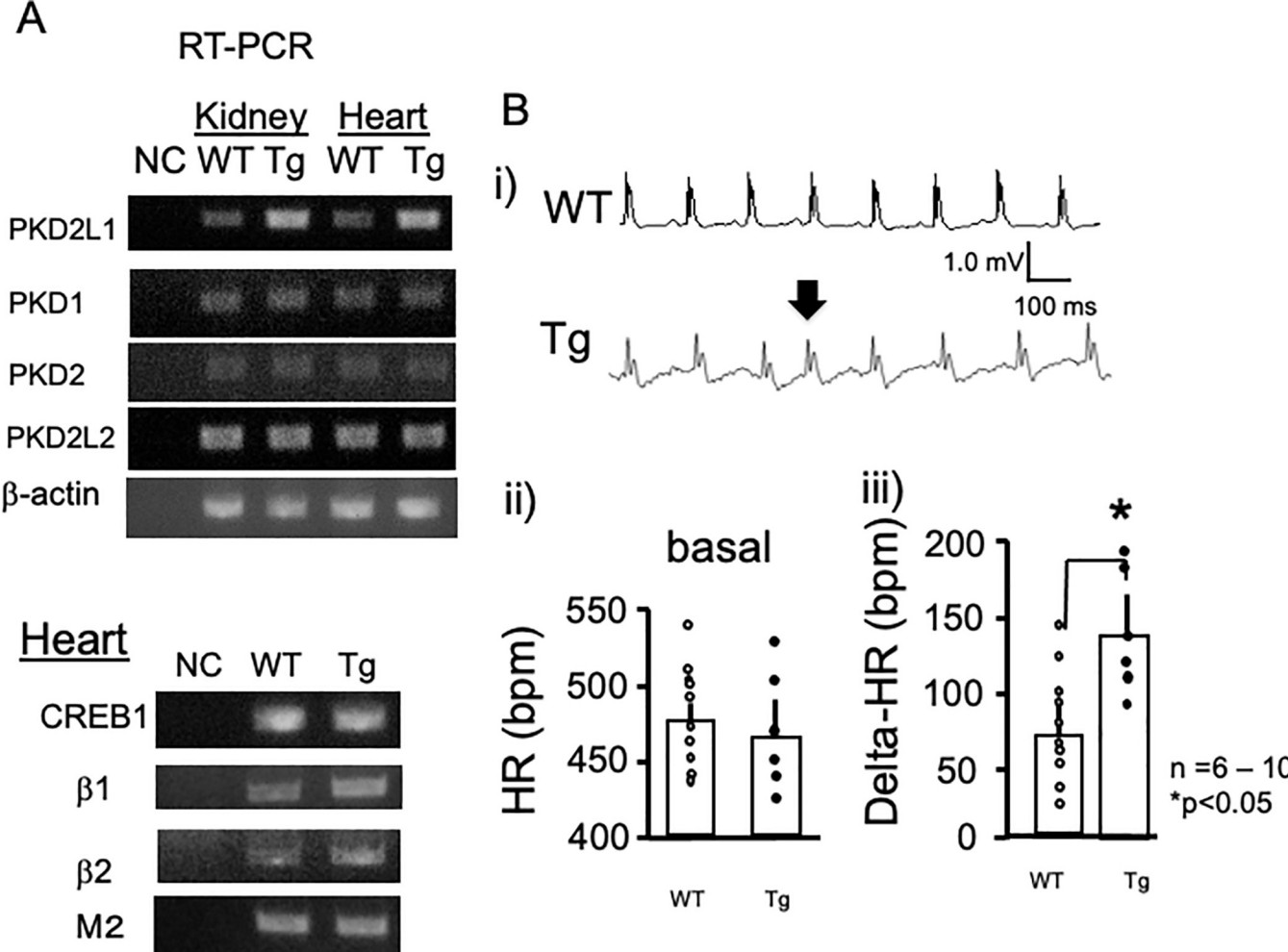

**Fig 2.** A) Typical RT-PCR data from kidney and heart tissues collected from WT and Tg mice. Amplified sequences are shown. Increased expression levels of PKD2L1 were confirmed in kidney and heart tissues from Tg mice. The expression levels of PKD1, PKD2, and PKD2L2 were assessed. Expression of β-actin was evaluated as a control. The primer sets used for PCR amplification are shown. B) Representative ECG tracings of WT and Tg mice under basal conditions (i). Statistical analysis of the heart rates of WT and Tg mice (ii). There were no significant differences in heart rate between WT (*n* = 10) and Tg (*n* = 6) mice under basal conditions. Isoproterenol administration resulted in increased heart rate changes in Tg mice (iii). *$P < 0.05$, between WT (n = 10) and Tg (n = 6) mice. Error bars indicate the standard error of the mean.

expression levels of the PKD1, PKD2, and PKD2L2 genes did not significantly differ between Tg and WT mice. Conversely, the expression levels of β1-adrenergic receptors and β2ARs were elevated (Fig 2, S3 Fig in S1 File), while CREB1 and muscarinic M2 receptor expression levels were not affected in the heart.

We also examined the expression levels of other genes involved in cardiac hypertrophy, but found no significant changes (S1 Fig in S1 File). Additionally, we analyzed RT-PCR with a primer set, which distinguished the endogenous PKD2L1 gene (507 bp) and the transgene (235 bp). The WT heart showed a single band, while Tg mice showed an additional 235-bp band, indicating expression of the transgene (S1 Fig in S1 File). Taken together, the results showed that overexpression of dominant-negative PKD2L1 resulted in enhanced expression of β-adrenergic receptors.

## Modified pacemaking in the heart

Previously, Yao et al. reported that PKD2L1 interacts and co-localizes with β2AR, a G-protein-coupled receptor active in cAMP production, on the primary cilia of neuronal cells in the brain. Therefore, we speculated that PKD2L1 might participate in a protein kinase A (PKA)-dependent pathway by interacting with β-adrenergic receptors in the heart. To evaluate this possibility, we performed ECG. Fig 2B shows representative ECG traces from WT control (upper panel) and PKD2L1del-Tg (lower panel) mice. Although most ECG results showed regular waves and rhythms, PKD2L1del-Tg mice demonstrated some supraventricular premature contraction (arrow). Because the PKA-dependent pathway is often associated with cAMP-related arrhythmia, our present data suggest the involvement of PKD2L1 in a PKA-dependent pathway. Statistical analysis revealed no significant differences in basal heart rate. However, enhanced responsiveness to isoproterenol (4 μg/kg), a β-adrenergic agonist, was observed in PKD2L1del-Tg mice (Fig 2B).

## Modified HRV

Because PKD2L1del-Tg mice showed enhanced responsiveness to isoproterenol, we performed ECG analysis to explore heart rate regulation in these mice using Power Lab HRV software. Fig 3A shows representative *Poincaré* plots for WT (left panel) and PKD2L1del-Tg mice (right panel). WT mice exhibited stable changes in R-R intervals, whereas PKD2L1del-Tg mice exhibited unstable R-R intervals (shown as dots in Fig 3A). These data suggested unstable pacemaking in the Tg mice, which coincided with supraventricular premature contraction on basal ECG (Fig 2B, arrow). Power spectral analysis (Fig 3B) showed no significant differences between the groups, whereas the standard deviation of SDNN was significantly greater in PKD2L1del-Tg mice (Fig 3C). This indicated unstable R-R intervals, suggestive of modified sympathetic nerve regulation in the Tg mice.

## Enhanced atrial contraction

Because of increased β-adrenergic receptors and enhanced responsiveness to isoproterenol in ECG, atrial contraction was analyzed. Sympathetic responsiveness was evaluated by measuring inotropic responses to isoproterenol in isolated spontaneously beating right atria. Fig 4A shows force changes in response to isoproterenol (100 nM) in the atria from WT and PKD2L1del-Tg mice. Isoproterenol caused dose-dependent changes in force within atria from WT and PKD2L1del-Tg mice; the responsiveness at high doses was enhanced in PKD2L1del-Tg compared with WT atria (Fig 4B). Norepinephrine and epinephrine (other catecholamines) elicited similar results, indicating that PKD2L1del-Tg mice possess enhanced responsiveness to β-adrenergic stimulation (S2 Fig in S1 File).

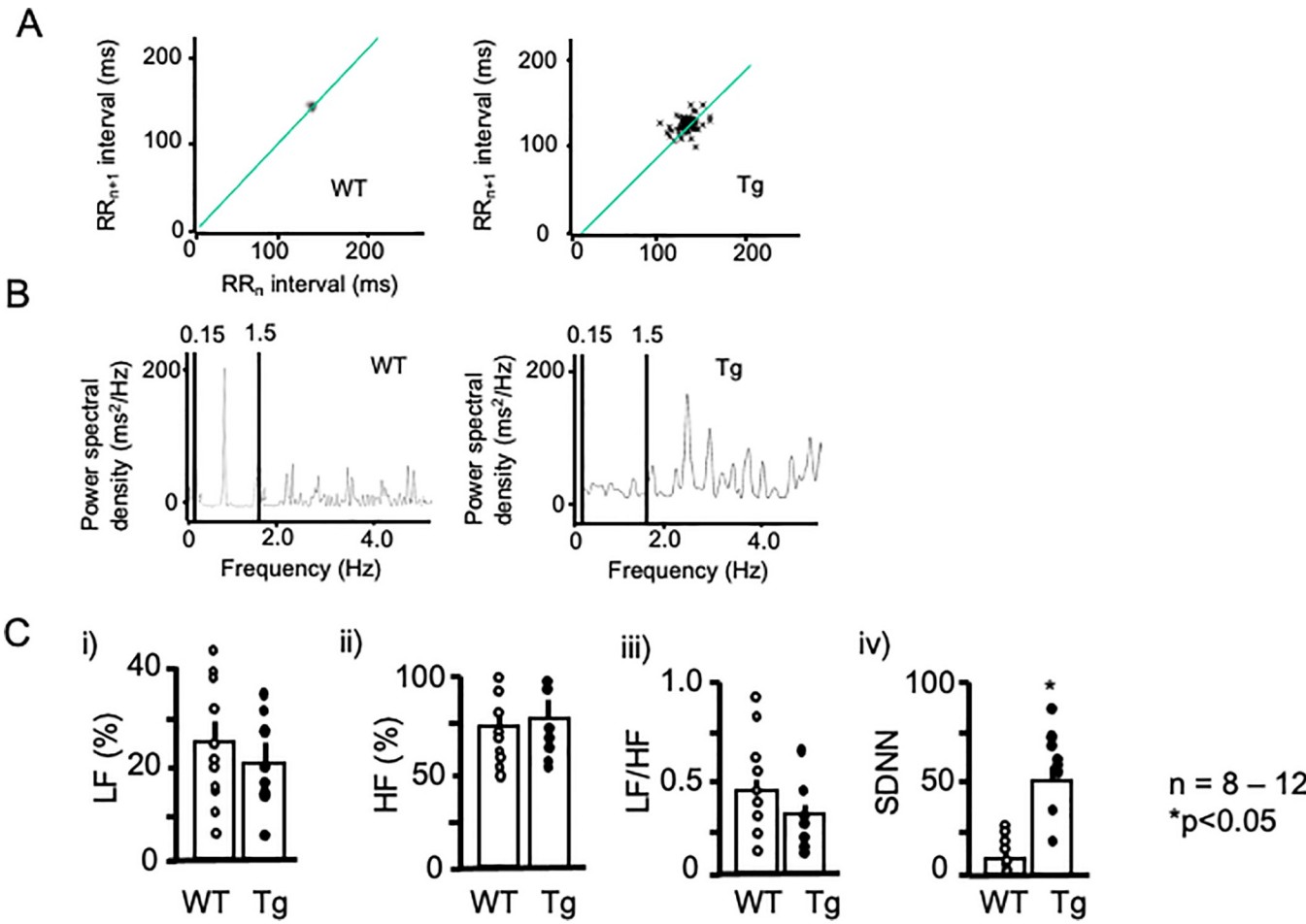

**Fig 3. HRV analysis.** A) Representative ECG *Poincaré* plots of WT (left panel) and Tg mice (right panel). *Poincaré* plots (RRn vs. RRn+1) in which consecutive pairs of R-R intervals during the control period were plotted with the nth+1 R-R interval against the nth R-R period. Note the marked scattering in Tg mice. B) Representative power spectral analysis of WT (left panel) and Tg mice (right panel). C) Statistical analysis of the power spectra (i–iii). LF (i) and HF (ii) components, LF/HF (iii), and SDNN of WT (open circles) and Tg mice (filled circles) are shown. $^*P < 0.05$ vs. WT. Each group consisted of at least six samples. Error bars indicate the standard error of the mean.

## Action potential changes

Because of the enhanced responsiveness to β-adrenergic stimulation in Tg atrial contraction, we examined atrial cardiac action potentials. Fig 5A shows representative traces of basal cardiac action potentials in the atria from a WT mouse (Fig 5A left upper panel) and a PKD2L1del-Tg mouse (right upper panel). The rapid upstroke of the cardiac action potential was followed by a plateau phase and subsequent rapid terminal repolarization. Isoproterenol (100 nM) shortened the action potential duration in both mice (lower panels). Statistical analysis revealed no significant effects on resting membrane potential or action potential amplitude, whereas the 90% action potential duration was significantly shortened in the atria of PKD2L1del-Tg mice (Fig 5A red arrow and 5B).

## Discussion

In this study, we established a mouse line overexpressing the dominant-negative form of PKD2L1, which cannot form a cation channel. The Tg mice showed enhanced expression of

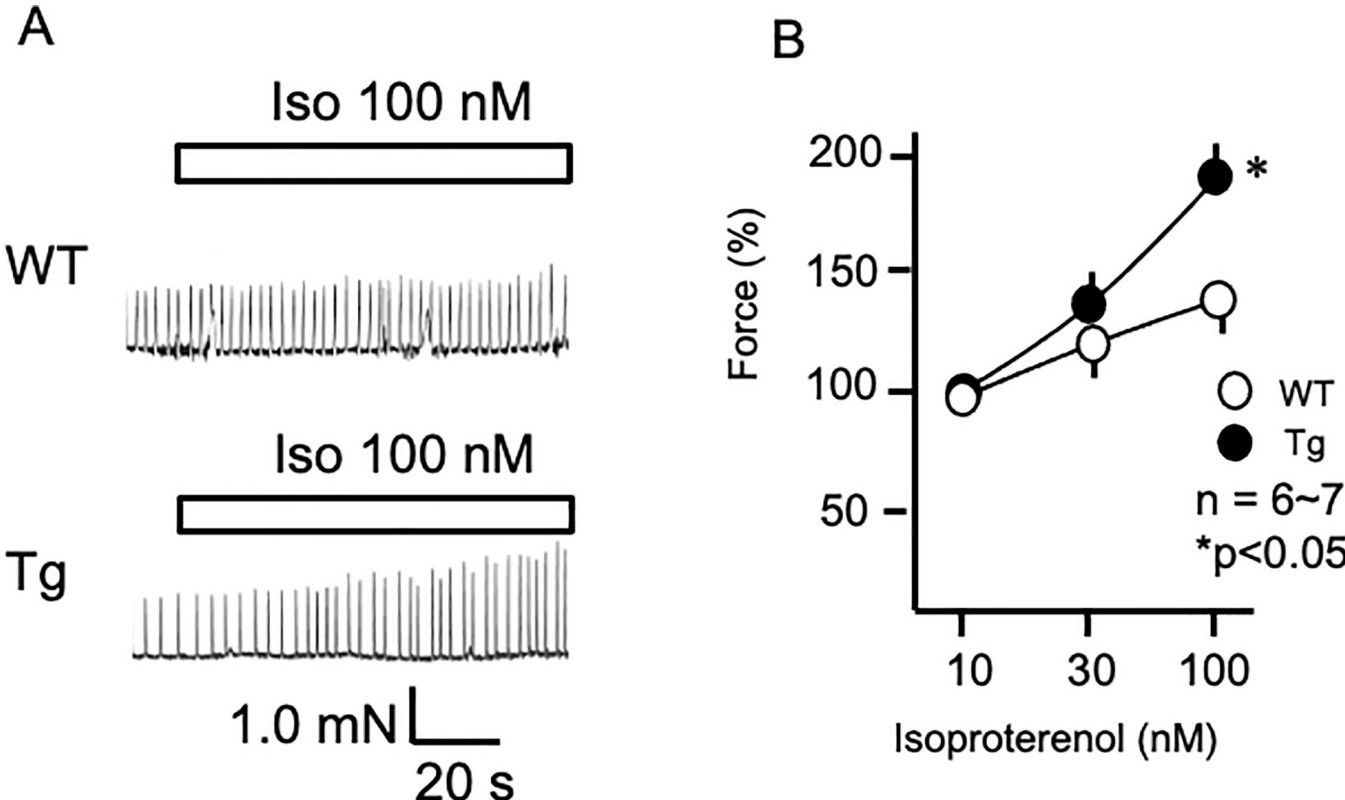

**Fig 4. Increased contractility in PKD2L1del-Tg atria in response to isoproterenol.** A) Typical traces of atrial contraction in WT (upper panels) and PKD2L1del-Tg (lower panels) mice. Isoproterenol (100 nM) increased atrial contractility in WT mice. Tg mice showed enhanced atrial contractility to isoproterenol. B) Dose-dependent changes in atrial contractility in response to isoproterenol (10–100 nM) in WT and Tg mice. Error bars indicate the standard error of the mean. $^*P < 0.05$, between atria from WT and Tg mice.

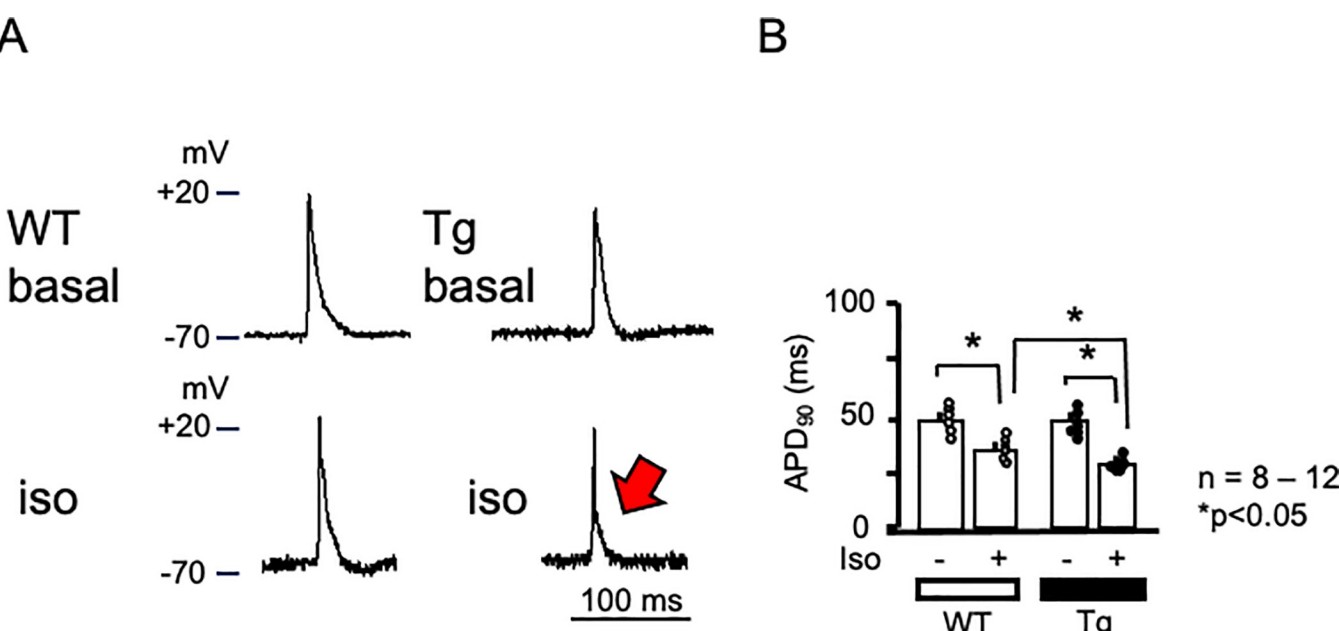

**Fig 5. Action potential in atria.** A) Representative action potentials of atrial sinus nodes in WT and PKD2L1del-Tg mice. Basal action potential changes in WT and Tg mice are shown (upper panels). Action potential changes are shown in response to isoproterenol (100 nM, lower panels). B) Statistical analysis of the atrial action potential duration at 90% repolarization in WT and Tg mice. Atria from Tg mice showed a shortened action potential duration (red arrow). $n = 8–12$. $P < 0.05$.

PKD2L1, but no significant changes in other PKD-related genes. Comparative RT-PCR analysis revealed elevated expression levels of the β1- and β2-adrenergic receptors in the hearts of PKD2L1del-Tg mice. The PKD2L1del-Tg mice sometimes showed supraventricular premature contraction. Furthermore, the PKD2L1del-Tg mice showed enhanced sensitivity to β-adrenergic stimulation on ECG. HRV analysis showed an unstable R-R interval and increased SDNN in PKD2L1del-Tg mice. Examination of spontaneous atrial contraction showed enhanced sensitivity to isoproterenol. Action potential recordings revealed a shortened action potential duration in response to isoproterenol in the atria of PKD2L1del-Tg-mice. Taken together, PKD2L1del-Tg mice showed enhanced adrenergic sensitivity, suggesting involvement of PKD2L1 in sympathetic regulation.

In a previous study, Yao et al. reported reduced cAMP and β-adrenergic responses in the brains of PKDL-deficient mice [12]. Consistent with their findings, our dominant-negative PKD2L1-overexpressing mice showed enhanced β-adrenergic responsiveness. Choi et al. reported the involvement of PKD2 in A-kinase anchoring protein complexes in kidney epithelial cells [25]. Considering the high similarity between PKD2 and PKD2L1 [10], the dominant-negative form of PKD2L1 might enhance the PKA cascade, although future studies are needed to confirm this hypothesis. Nevertheless, the findings in our study suggest a close relationship between PKD-forming channels and the cAMP cascade.

Lu et al. reported that high-salt-induced cardiac hypertrophy occurs in PKD2L1-deficient mice [14]. Similar to PKD2L1, transient receptor potential channels participate in the pathogenesis of cardiovascular disease [26]. PKDD channels, including PKD2, PKD2L1, and PKD2L2, are nonselective ion channel proteins associated with ADPKD [27]. Cardiac hypertrophy is an important clinical complication in patients with ADPKD [28].

Yao et al. reported interactions between PKD2L1 and β2AR, both *in vitro* and *in vivo* [12]. PKD2L1 deficiency did not affect the protein level of β2AR in the brain but caused loss of ciliary β2AR in neurons, suggesting that PKD2L1 is required for β2AR ciliary localization. In the study by Yao et al., PKD2L1-null mice showed reduced cAMP levels in the brain. Furthermore, the PKD2 channel, which demonstrates a high degree of homology with PKD2L1, participates in cAMP homeostasis in a primary cilium-dependent manner [25]. Additional analyses are needed to elucidate the precise mechanism of β2AR trafficking and the roles of PKD channels in controlling the cAMP signaling cascade. In the present study, overexpression of the dominant-negative form of PKD2L1 resulted in enhanced expression of β1-adrenergic receptors and β2ARs in the heart (S3 Fig in S1 File). In addition, PKD2L1del-Tg mice showed increased responsiveness to isoproterenol. Thus, we speculate that overexpression of PKD2L1 induces transcription of β1- and β2-adrenergic receptors. Additionally, overexpressed PKD2L1 might interact with β1- or β2-adrenergic receptors, resulting in enhanced sensitivity to isoproterenol.

To determine the mechanisms underlying enhanced sympathetic responses in PKD2L1del-Tg mice, the interaction between PKD2L1 and β-adrenergic receptors should be evaluated in future studies. In addition, the levels of cAMP and PKA-dependent phosphorylation of target molecules, such as CREB, should be analyzed. The PKD2L1 ionic currents in cardiac myocytes from wild-type and PKD2L1del-transgenic mouse should also be evaluated.

In conclusion, we established a Tg mouse line overexpressing a dominant-negative form of PKD2L1. The novel Tg mice showed enhanced sensitivity to β-adrenergic stimulation in the ECG analysis. Examination of HRV showed an unstable R-R interval and increased SDNN in PKD2L1del-Tg mice, suggesting modified sympathetic nerve regulation. Spontaneous atrial contraction and action potential recordings demonstrated enhanced sensitivity to isoproterenol in the atria of PKD2L1del-Tg mice. Taken together, our findings strongly suggest that PKD2L1 is involved in sympathetic regulation.

## Supporting information

**S1 File.**
(DOCX)

**S1 Raw images.**
(TIFF)

## Acknowledgments

We thank Maximilian Murakami for his technical assistance.

## Author Contributions

**Conceptualization:** Manabu Murakami.

**Formal analysis:** Manabu Murakami, Hirofumi Tomita.

**Funding acquisition:** Manabu Murakami, Kazuyoshi Hirota, Shirou Itagaki.

**Investigation:** Manabu Murakami, Agnieszka M. Murakami, Takayuki Nemoto, Takayoshi Ohba, Manabu Yonekura, Yuichi Toyama, Yujiro Asada, Ichiro Miyoshi.

**Methodology:** Manabu Yonekura, Yasushi Matsuzaki, Daisuke Sawamura, Kazuyoshi Hirota.

**Resources:** Yasushi Matsuzaki, Daisuke Sawamura.

**Visualization:** Hirofumi Tomita.

**Writing – original draft:** Manabu Murakami.

**Writing – review & editing:** Manabu Murakami, Hirofumi Tomita, Shirou Itagaki.

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
