## [Decision Letter · Decision Letter 0]

29 Jun 2021

PONE-D-21-08901

Enhanced β-adrenergic response in mice with dominant-negative expression of the PKD2L1 channel

PLOS ONE

Dear Dr. Murakami,

Thank you for submitting your manuscript to PLOS ONE. After careful consideration, we feel that it has merit but does not fully meet PLOS ONE’s publication criteria as it currently stands. Therefore, we invite you to submit a revised version of the manuscript that addresses the points raised during the review process. While addressing the reviewer's concerns, please better uncropped images of Western blots. Also, better PCR images are needed. All bar graphs should be converted to dot plots with indicated mean and error bars.

We look forward to receiving your revised manuscript.

Kind regards,

Alexander G Obukhov, Ph.D.

Academic Editor

PLOS ONE

Journal Requirements:

2. To comply with PLOS ONE submissions requirements, please provide methods of sacrifice in the Methods section of your manuscript. 

This research was sponsored in part by Grants-in-Aid for Scientific Research from JSPS, KAKENHI (17K08527, 17H04319, 16K09489, and 20K07255).

This research was sponsored in part by Grants-in-Aid for Scientific Research from JSPS, KAKENHI (17K08527, 17H04319, 16K09489, and 20K07255).

6. Please upload a copy of Supporting Information Figures S1, S2, S3 and S4 which you refer to in your text on page 24 and 25.

Reviewers' comments:

Reviewer's Responses to Questions

**Comments to the Author**

1. Is the manuscript technically sound, and do the data support the conclusions?

Reviewer #1: Yes

2. Has the statistical analysis been performed appropriately and rigorously? 

Reviewer #1: Yes

3. Have the authors made all data underlying the findings in their manuscript fully available?

Reviewer #1: Yes

4. Is the manuscript presented in an intelligible fashion and written in standard English?

Reviewer #1: Yes

5. Review Comments to the Author

Reviewer #1: Polycystic kidney disease 2-like 1 protein also known as transient receptor potential polycystic 2 (TRPP2) is a protein that in humans is encoded by the PKD2L1 gene. This study reports the development and characterisation of a transgenic mouse line overexpressing a dominant negative form of PKD2L1. The phenotype showed an augmented sensitivity to B-adrenergic stimulation, an unstable R-R interval and elevated paramertes of sympathetic regulation. The paper is generally well written and presented, and the data will be of interest to better understand the pathogenesis of ADPKD.

Comments

1. The Introduction seems dis-jointed. For example, the rationale and link with ADPKD is not clear and could be better explained

2. As this is the first description of this mouse line, it would be important to report whether there any other phenotype, such as the histological presence of kidney cysts and/or vascular abnormalities

6. PLOS authors have the option to publish the peer review history of their article (what does this mean?). If published, this will include your full peer review and any attached files.

Reviewer #1: No

---

## [Author Response · Author response to Decision Letter 0]

15 Jul 2021

My coauthors and I wish to submit our manuscript, entitled “Enhanced β-adrenergic response in mice with dominant-negative expression of the PKD2L1 channel,” for consideration for publication as an Article in PLoS ONE.

We thank the editor and reviewer for reviewing our manuscript and for the suggested revisions. We have addressed all of the reviewers’ queries. 

The authors have ensured that the revised manuscript conforms to the style requirements of PLOS ONE. The authors have converted the bar graphs into bar graphs (means ± standard errors) with dots. 

The authors have provided the original uncropped and unadjusted images of DNA typing and reverse-transcription PCR in Supporting Information (SI) Fig. 4A. The original whole western blot images in Fig. 1C are provided in SI Fig. 4B. 

The authors have corrected the funding information in accordance with the suggestion.

The authors have provided a minimal data set in the SI according to the suggestion.

The Poincaré and power spectral density plots in Fig. 2A and B were calculated from one mouse electrocardiogram recording. A statistical analysis of the electrocardiogram recordings was performed for the heart rate variability analysis using the power spectrum low frequency (LF) and high frequency (HF), LF/HF, and normal-to-normal R-R intervals, which were calculated from the electrocardiogram recordings (Fig. 3C).

Our point-by-point responses to each of the comments are presented in the response letter.

Sincerely,

Manabu Murakami

Department of Pharmacology, 

Hirosaki University, Graduate School of Medicine

E-mail: mmura0123@hotmail.co.jp

---

## [Decision Letter · Decision Letter 1]

20 Oct 2021

PONE-D-21-08901R1Enhanced β-adrenergic response in mice with dominant-negative expression of the PKD2L1 channelPLOS ONE

Dear Dr. Murakami,

Thank you for submitting your manuscript to PLOS ONE. After careful consideration, we feel that it has merit but requires some minor revisions before it can be accepted for publication in PLOS ONE. Therefore, we invite you to submit a revised version of the manuscript that addresses the points raised by the reviewer.

We look forward to receiving your revised manuscript.

Kind regards,

Alexander G Obukhov, Ph.D.

Academic Editor

PLOS ONE

Journal Requirements:

Reviewers' comments:

Reviewer's Responses to Questions

**Comments to the Author**

1. If the authors have adequately addressed your comments raised in a previous round of review and you feel that this manuscript is now acceptable for publication, you may indicate that here to bypass the “Comments to the Author” section, enter your conflict of interest statement in the “Confidential to Editor” section, and submit your "Accept" recommendation.

Reviewer #2: All comments have been addressed

2. Is the manuscript technically sound, and do the data support the conclusions?

Reviewer #2: Yes

3. Has the statistical analysis been performed appropriately and rigorously? 

Reviewer #2: I Don't Know

4. Have the authors made all data underlying the findings in their manuscript fully available?

Reviewer #2: Yes

5. Is the manuscript presented in an intelligible fashion and written in standard English?

Reviewer #2: Yes

6. Review Comments to the Author

Reviewer #2: Murakami et al. created PKD2L1del-Tg mice to investigate physiological role of PKD2L1channel, particularly for β-adrenergic stimulation. Using this new model, the authors measured EKG, inotropic atrial contractions, and action potentials. Their measurements show that PKD2L1del-Tg mice have increased adrenergic sensitivity indicating PKD2L1 involvement to sympathetic regulation. The study is important and well-done. I only have a question about statistics and an optional suggestion to improve the paper Discussion.

Please clarify statistics:

Figures 4 and 5: What is “n” in the figure legends: n =6~7 and n = 8-12. Is it number of animals, number of cells, or number of measurements?

Discussion:

PKD2L1 is a cation channel. However according to the literature and results of the present study, its involvement to sympathetic regulation may not be necessarily linked to its activity as a channel, but to its role in regulation of β2AR trafficking or some other activity. The paper discussion is rather short, and I would encourage the authors to write more about possible mechanisms of PKD2L1 involvement in sympathetic regulation. Based on the new results of the present study and literature data, please make a schematic figure to illustrate possible interactions and possible mechanisms of how PKD2L1 regulate β-adrenergic stimulation.

Line :370: “Additional analyses are needed to elucidate the precise mechanism of β2AR trafficking and the roles of PKD channels in controlling the cAMP signaling cascade.”

Please specify what specific analyses and/or experiments are needed to clarify the mechanisms.

7. PLOS authors have the option to publish the peer review history of their article (what does this mean?). If published, this will include your full peer review and any attached files.

Reviewer #2: No

---

## [Author Response · Author response to Decision Letter 1]

29 Nov 2021

We thank the editor and reviewers for reviewing our manuscript and suggesting revisions. We have addressed all of the reviewers’ queries. 

Our point-by-point responses to each of the comments are presented in the rebuttal letter (Our responses are in red).

Sincerely yours,

Manabu Murakami

---

## [Decision Letter · Decision Letter 2]

9 Dec 2021

Enhanced β-adrenergic response in mice with dominant-negative expression of the PKD2L1 channel

PONE-D-21-08901R2

Dear Dr. Murakami,

We’re pleased to inform you that your manuscript has been judged scientifically suitable for publication and will be formally accepted for publication once it meets all outstanding technical requirements.

Kind regards,

Alexander G Obukhov, Ph.D.

Academic Editor

PLOS ONE

Reviewers' comments:

Reviewer's Responses to Questions

**Comments to the Author**

1. If the authors have adequately addressed your comments raised in a previous round of review and you feel that this manuscript is now acceptable for publication, you may indicate that here to bypass the “Comments to the Author” section, enter your conflict of interest statement in the “Confidential to Editor” section, and submit your "Accept" recommendation.

Reviewer #2: All comments have been addressed

2. Is the manuscript technically sound, and do the data support the conclusions?

Reviewer #2: Yes

3. Has the statistical analysis been performed appropriately and rigorously? 

Reviewer #2: Yes

4. Have the authors made all data underlying the findings in their manuscript fully available?

Reviewer #2: Yes

5. Is the manuscript presented in an intelligible fashion and written in standard English?

Reviewer #2: Yes

6. Review Comments to the Author

Reviewer #2: The authors have satisfactorily addressed my comments. The paper has been improved. I have no further comments.

7. PLOS authors have the option to publish the peer review history of their article (what does this mean?). If published, this will include your full peer review and any attached files.

Reviewer #2: No

---

## [Editor Report · Acceptance letter]

7 Jan 2022

PONE-D-21-08901R2 

Enhanced β-adrenergic response in mice with dominant-negative expression of the PKD2L1 channel 

Dear Dr. Murakami:

I'm pleased to inform you that your manuscript has been deemed suitable for publication in PLOS ONE. Congratulations! Your manuscript is now with our production department. 

Kind regards, 

on behalf of

Dr. Alexander G Obukhov 

Academic Editor

PLOS ONE